# Whole genome case-control study of central nervous system toxicity due to antimicrobial drugs

Joel Ås[1], Ilma Bertulyte[1], Nina Norgren[2], Anna Johansson[3], Niclas Eriksson[1,4], Henrik Green[5,6], Mia Wadelius[1]*, Pär Hallberg[1]

1 Department of Medical Sciences, Clinical Pharmacogenomics and Science for Life Laboratory, Uppsala University, Uppsala, Sweden, 2 Department of Molecular Biology, National Bioinformatics Infrastructure Sweden, Science for Life Laboratory, Umeå University, Umeå, Sweden, 3 Dept of Cell and Molecular Biology, National Bioinformatics Infrastructure Sweden, Science for Life Laboratory, Uppsala University, Uppsala, Sweden, 4 Uppsala Clinical Research Center, Uppsala, Sweden, 5 Division of Clinical Chemistry and Pharmacology, Department of Biomedical and Clinical Sciences, Linköping University, Linköping, Sweden, 6 Department of Forensic Genetics and Forensic Toxicology, National Board of Forensic Medicine, Linköping, Sweden

* mia.wadelius@medsci.uu.se

**Citation:** Ås J, Bertulyte I, Norgren N, Johansson A, Eriksson N, Green H, et al. (2024) Whole genome case-control study of central nervous system toxicity due to antimicrobial drugs. PLoS ONE 19(2): e0299075. https://doi.org/10.1371/journal.pone.0299075

**Data Availability Statement:** Full processing and analysis pipeline can be found at https://github.com/JoelAAs/WGS_pipeline, additional downstream analysis and plotting can be found at:

## Abstract

A genetic predisposition to central nervous system (CNS) toxicity induced by antimicrobial drugs (antibiotics, antivirals, antifungals, and antiparasitic drugs) has been suspected. Whole genome sequencing of 66 cases and 833 controls was performed to investigate whether antimicrobial drug-induced CNS toxicity was associated with genetic variation. The primary objective was to test whether antimicrobial-induced CNS toxicity was associated with seventeen efflux transporters at the blood-brain barrier. In this study, variants or structural elements in efflux transporters were not significantly associated with CNS toxicity. Secondary objectives were to test whether antimicrobial-induced CNS toxicity was associated with genes over the whole genome, with HLA, or with structural genetic variation. Uncommon variants in and close to three genes were significantly associated with CNS toxicity according to a sequence kernel association test combined with an optimal unified test (SKAT-O). These genes were LCP1 (q = 0.013), RETSAT (q = 0.013) and SFMBT2 (q = 0.035). Two variants were driving the LCP1 association: rs6561297 (p = $1.15 \times 10^{-6}$, OR: 4.60 [95% CI: 2.51–8.46]) and the regulatory variant rs10492451 (p = $1.15 \times 10^{-6}$, OR: 4.60 [95% CI: 2.51–8.46]). No common genetic variant, HLA-type or structural variation was associated with CNS toxicity. In conclusion, CNS toxicity due to antimicrobial drugs was associated with uncommon variants in LCP1, RETSAT and SFMBT2.

## 1. Introduction

Antimicrobial drugs (antibiotics, antivirals, antifungals and antiparasitic drugs) are associated with central nervous system (CNS) toxicity in some patients [1–3]. Well-known examples are ototoxicity due to aminoglycosides, encephalopathy and seizures caused by beta-lactam

https://github.com/JoelAAs/WGS_testing Genetic data will stored at the official Federated European Genome-phenome Archive (FEGA) at NBIS/Elixir (https://fega.nbis.se/). Data cannot be shared publicly because of patient agreement and GDPR. Data can be requested from the Swedegene data access committee: info@swedegene.se.

**Funding:** This study was granted by the Science for Life Laboratory's Swedish Genomes Program 2017 that was supported by the Knut and Alice Wallenberg Foundation (application ID NP:00085). Furthermore, we received grants from the Swedish Research Council (Medicine 521-2011-2440, 521-2014-3370 and 2018-03307), and Clinical Research Support (Avtal om Läkarutbildning och Forskning, ALF) at Uppsala University. All grants mentioned were granted to MW and PH. The funders had no role in study design, data collection and analysis, decision to publish, or preparation of the manuscript.

**Competing interests:** The authors declare no conflict of interest. No funder took part in the study design, recruitment, analysis, interpretation of data, writing of the report or in the decision to submit the paper for publication. There are no other conflicts of interest. Genetic data will be available at the official Federated European Genome-phenome Archive (FEGA) at NBIS/Elixir (https://fega.nbis.se/). Data access can be requested if the recipient adheres to our ethics approval, and the request is in accordance with FAIR data management and the General Data Protection Regulation (GDPR) 2016/679.

antibiotics, benign intracranial hypertension due to tetracyclines, neuropsychiatric effects of trimethoprim-sulfamethoxazole, quinolones, antimalarials and antivirals, and cerebellar toxicity due to metronidazole. The risk of CNS toxicity is believed to be influenced by pharmacokinetic and genetic variation and by the penetration of drugs across the blood-brain barrier [4]. Efflux transporters play an important role for the exclusion of antimicrobial drugs from the brain. The efflux transporters P-gp (encoded by ABCB1) and BCRP (encoded by ABCG2) that are highly expressed in the blood-brain barrier are known to transport a number of antimicrobial drugs from the apical membrane or cytoplasm back into the bloodstream.

Genetic variation in transporter genes has been associated with changes in pharmacokinetics and with certain adverse drug reactions (ADRs) [5]. However, few studies have described associations between polymorphisms in transporter genes and the risk of antimicrobial-induced CNS toxicity. Some data support a role for ABCB1 variants in neuropsychiatric ADRs of mefloquine [6–8]. The mechanism behind serious neurological ADRs of ivermectin has been debated, largely focusing on the role of concomitant infection versus the presence of ABCB1 variants allowing penetration into the CNS. ABCC2 and ABCG2 polymorphisms and blood-brain barrier permeability have further been shown to affect concentrations of ceftriaxone in the cerebrospinal fluid [6–9].

To the best of our knowledge, no study has systematically investigated whether genetic variation in drug transporters could predispose to CNS toxicity associated with antimicrobial drugs. The primary aim of this study was to investigate whether genetic variation in transporters expressed at the blood-brain barrier is associated with increased risk of CNS toxicity of antimicrobials. A secondary aim was to explore whether genetic variation across the whole genome was associated with risk of CNS toxicity.

## 2. Materials and methods

This is case-control study of possible genetic variation underlying CNS toxicity to antimicrobial drug. All cases exhibited CNS-toxic adverse reactions to antimicrobial drug treatment. Table 2 provides a comprehensive list of the drugs and adverse reactions, while section 3.1 presents detailed patient characteristics. Genetic information was obtained through whole genome sequencing.

### 2.1 Ethics statement

The study was conducted according to the guidelines of the Declaration of Helsinki, and the study was approved by the regional ethical review board in Uppsala, Sweden (2010/231 Uppsala). Written informed consent was obtained from all participants in the study.

### 2.2 Code availability

Full preprocessing pipeline and downstream analysis code is available at github, and urls are found under data availability statement.

### 2.3 Sample description

Patients were identified using the SWEDEGENE method [10]. In brief, patients who had been reported to the Medical Products Agency's national register of ADRs due to a suspected CNS-toxic adverse reaction 1990–2017 were retrospectively contacted by SWEDEGENE and recruited July 2010—August 2017.

All patients provided informed consent and were at least 18 years of age at the time of recruitment. Clinical data (demographics, medical history, drug treatment history, laboratory

data, and ancestry) were collected through interviews using a standardized questionnaire, and by obtaining and reviewing medical records. The authors had access to information that could identify individual participants during data collection.

Blood samples were drawn at the patient's nearest health-care facility 2010–2017 and were sent to SWEDEGENE where they were stored at -70˚C. This case-control study used data from whole genome sequencing (WGS) that was performed in 2018. The cases were patients who had experienced CNS toxicity after initiation of an antimicrobial drug (anatomical therapeutic classes [ATC] J01, J02, J04, J05, J06, P, L03AB11). Each case was adjudicated by two specialists in clinical pharmacology and drug safety. The controls constituted sequenced patients who had experienced other forms of ADRs from a multitude of drugs.

## 2.4 Library preparation and sequencing

DNA extraction was performed by the Karolinska Institute biobank using the Chemagen kit with an input of 400 μL peripheral blood for each sample. Library preparation and sequencing were performed by the National Genomics Infrastructure (NGI) at the SNP&SEQ Technology Platform in Uppsala (NGI-U), Sweden. Sequencing libraries were prepared from 1 μg DNA using the TrueSeq PCRfree DNA sample preparation kit, targeting an insert size of 350 bp. The library preparation was performed according to the manufacturer's instructions. Pair-end sequencing with read length of 150 bp was performed on Illumina HiSeq X using v2.5 Sequencing chemistry to a target average of 30x coverage. The full cohort was sequenced in four batches of 704, 262, 10 and 2 samples. The two last batches consisted of samples resequenced due to insufficient sequence quality or depth determined by NGI-U. The cases were spread randomly throughout the batches. The 79 samples not included in this study were excluded due to being evaluated as uncertain cases, being part of a study with a strong known association to HLA-type (narcolepsy) or outliers in genomic PCA. In the PCA plot (S1 Fig) only the included cases and controls are shown.

## 2.5 Alignment, recalibration and PCR deduplication

Alignment, base quality recalibration and marking of PCR duplication were performed by NGI-U. In brief, sample reads were aligned to Genome Reference Consortium Human Build 37 (GRCh37) using BWA-MEM 0.7.12 [11] and indexed using samtools 0.1.19 [12]. Subsequent alignments were locally realigned around structural variation and indels using Genome Analysis Toolkit (GATK) 3.3 RealignerTargetCreator and GATK 3.3 IndelRealigner [13]. Picard MarkDuplicates 1.120 were used for PCR deduplication and base quality recalibration tables were generated using GATK 3.3 BaseRecalibrator. Alignment quality control statistics and sequencing depth plots were gathered tested and using samtools stats, BamQC, King AutoQC and BcfTools stats. These results was inspected using Qualimap v2.2 [14] and multiQC and screened for any outlying regions and samples, however, none were found.

## 2.6 Variant calling and joint genotyping

Following the best practice GATK workflow [15], per sample genomic variant calling files (gVCFs) were called using GATK 3.8 HaplotypeCaller [16]. Single-sample files were conjoined using GATK CombineGVCFs into 10 Mb subsections. Genotyping was performed on each of the subsections using GATK GenotypeGVCF. Subsequently, all subsections were concatenated using VCF-tools, and the variants in the full VCF were recalibrated using GATK Variant Quality Score Recalibration (VQSR). VQSR was executed twice sequentially, once for single nucleotide polymorphisms (SNPs) and once for insertions/deletions (indels).

## 2.7 Filtering

Multiallelic SNPs and indels were split using BCFtools [17]. VCFtools was used to filter out variants that failed quality criteria marked previously in filtering steps, such variants failed by VQSR. Variant based missingness was calculated using KING autoQC [18] and any variant with over 5% missing calls was filtered out. Hardy-Weinberg equilibrium was calculated on the controls using PLINK2.0 and any variant with a p-value below $10^{-8}$ was filtered out. This resulted in a multi-sample VCF containing 45 million variants in 899 individuals.

## 2.8 Structural variation

Structural variation was called using the FindSV pipeline (https://github.com/J35P312/FindSV). FindSV takes single bam files as input and performs structural variant calling by TIDDIT [19] and CNVnator [20]. TIDDIT identifies chromosomal rearrangements by clustering contigs, split-reads and discordant pairs (paired reads that do not adhere to insert length), while CNVnator estimates any divergence in copy number along the chromosomes using a mean-shift method for coverage. The resulting single sample variant calling files (VCFs) were merged into a multisample VCF using VCFtools.

## 2.9 Annotation

Variant annotation as well as gene-based annotation was done using ANNOVAR [21]. Variant effect prediction was estimated using SnpEff [22] and VEP [23].

## 2.10 Principal component analysis

Principal component analysis (PCA) was performed to explore the genetic diversity and identify outliers within the dataset (S1 Fig). Prior to PCA, all variants without dbSNP build 138 [24] annotations or any non-autosomal variation were excluded. The filtered dataset was merged with HapMap [25] r23a using PLINK1.9 [26]. Variants with a minor allele frequency < 0.01 or genotyping rate < 0.1 were filtered out, which resulted in 2949306 variants. Filtering, merging and PCA was performed using PLINK2.0. The individual's parents' country of birth was used to validate their position in the PCA plot.

## 2.11 HLA imputation

Human leukocyte antigen (HLA) haplotypes were called up to six-digits codes using the HLA TWIN program by Omixon.

## 2.12 Analysis setup

All analyses except for HLA association were conducted in accordance with the primary aim of the candidate genes first, and subsequently the secondary aim, the whole genome.

**2.12.1 Common variant association.** Variant associations were calculated in PLINK2.0 using logistic regression with firth-fallback and four principal components as covariates. The variant frequency (VF) threshold was the expected value of one homozygous individual among cases, which with 66 cases equates a threshold of $1/\sqrt{66} = 0.123$. This stringent VF threshold was set due to the small number of cases, and therefore the high effect needed for significant results, and in favor of testing less frequent variants using variant aggregation methods. Additionally, any variants that fell within a region masked byReapeatMasker [27] for GRCh37 were excluded.

*2.12.1.1 Primary aim*: *Candidate gene approach*. The selection of genes encoding drug transporters expressed within the CNS was based on Geier et al, and included a set of 17 genes

**Table 1. Genes encoding drug transporters expressed in the central nervous system [28].**

| Gene | Protein |
|---|---|
| ABCB1 | P-gp/MDR1 |
| ABCC5 | MRP5 |
| ABCG2 | BCRP |
| ABCC1 | MRP1 |
| ABCC3 | MRP3 |
| ABCC6 | MRP6 |
| ABCC4 | MRP4 |
| SLC22A5 | OCTN2 |
| SLCO1A2 | OATP1A2 |
| SLC15A1 | HPECT1/HPEPT1/PEPT1 |
| SLC19A1 | Folate transporter 1 |
| SLC22A3 | OCT3 |
| SLC22A6 | OAT1 |
| SLC47A1 | Multidrug and toxin extrusion protein 1 (MATE1) |
| SLC47A2 | Multidrug and toxin extrusion protein 1 (MATE2) |
| SLCO2B1 | OATP2B1 |
| SLC10A1 | Na+-taurocholate cotransporting polypeptide (NTCP) |

(Table 1) [28]. The association threshold was set using a Bonferroni correction for the number of variants that passed filtration, i.e., 0.05/N.

*2.12.1.2 Secondary aim*: *Whole genome approach*. Other genetic variants were processed in the same way as in the candidate gene approach but the significance threshold was set to the genome wide significance $p = 5 \times 10^{-8}$.

**2.12.2 Gene association with SKAT-O.** Gene association tests were conducted on variants with an observed VF below 0.123 (i.e. the variants that were filtered out from the common variant analysis) while adhering to the filtering criteria listed below. The tests were performed using a sequence kernel association test combined with an optimal unified test (SKAT-O) [29] with a linear weighted kernel. Weights were set as a sigmoidal taper dependent on the distance from the closest exon, 5' untranslated region (UTR) or 3'UTR. Parameters were set so that a weight of zero was obtained at a distance of 140 bases, effectively filtering out anything that is further away. The weighting criteria was set to filter out the numerous deep intronic variants, while keeping variation possibly affecting splicing. In addition, any genes with less than 3 distinct variants detected among cases or controls, or with a sum of carrier/non-carrier difference of less than 8, were excluded in order to avoid singular systems.

For the primary aim, the significance threshold was adjusted according to Bonferroni correction, while for the secondary aim, false discovery rate (FDR) corrected q-values were calculated [30–34]. A q-value below 0.05 was considered significant.

**2.12.3 Structural variation association.** Any structural variation was encoded on a dominant gene basis. If a larger structural variation was present in a gene, that patient was labeled as a carrier. Tests for association were performed with logistic regression on a gene basis, and genes with less than 7 carriers among cases and controls were excluded. Similarly, to the gene association tests with SKAT-O, the significance threshold was adjusted with Bonferroni correction.

**2.12.4 HLA association.** HLA-association was tested implementing a dominant gene model with logistic regression. All HLA haplotypes with 4-digit codes were tested with principal components one to four as covariates. The significance threshold was adjusted with Bonferroni correction.

# 3. Results

## 3.1 Patient characteristics

A total of 76 patients with suspected CNS toxicity due to antimicrobial drugs were available within the study cohort. Following adjudication, 10 patients were excluded as their symptoms were suspected to be due to peripheral neurotoxicity. Characteristics of the 66 included cases are shown in Table 2. There were more women than men (72% women). The parents' country of birth was taken from standardized questionnaires, and was used as a proxy for genetic origin. For the majority of the cases, both parents were born in Sweden (n = 55). Nine cases had parents born in other Nordic or central European countries. The remaining two had parents born in the Middle East and North America. Among the 833 controls, 686 (~82%) had both parents born in Sweden, and 44 had one parent from Sweden. Among those with both parents born abroad,32 had at least one parent from another Nordic country. The remaining 46 parents' place of birth was mainly in Europe, but other continents were represented as well. Twenty-five people had missing data on the parents' place of birth.

The cases were on 33 different suspected drugs, the most common being mefloquine. The most common treatment indication was malaria prophylaxis, and psychiatric disorders were the most frequently reported ADRs. A total of 32 cases did not have concomitant diseases in addition to the indication for treatment. In all cases, symptoms developed after starting therapy. In 64 of the 66 cases, therapy was stopped due to the ADRs. In 47 of these 64 cases, symptoms resolved or improved after withdrawal, and in the remaining cases there was permanent disability. These cases concerned disturbance or loss of taste or smell, facial paralysis or ototoxicity. In two cases, the treatment was continued as it was deemed too important to be withdrawn.

A literature review was conducted relating to evidence for the role of ABC and SLC transporters in the handling of the suspected drugs (S1 Table). Of the suspected drugs, 55% had data indicating that they were substrates for a transporter expressed within the blood-brain barrier. For some drugs, data were not available.

## 3.2 Common variant association

**3.2.1 Primary aim: Candidate gene approach.** Among the candidate genes, no significant association was found between CNS toxicity and selected common variants (p<0.00047), in the gene association analysis, or among structural elements (S2–S4 Figs).

**3.2.2 Secondary Aim: Whole genome approach.** In the exploratory analysis of CNS toxicity and common genetic variants (Fig 1), no single variant was below the genome wide significance threshold ($p < 5 \times 10^{-8}$). The 50 variants with the lowest p-values are shown in S2 Table.

## 3.3 Gene association with SKAT-O

Uncommon variants (VF < 0.123) in 19250 genes were tested using optimal kernel machine methods and q-values were calculated.

Three genes had a q-value below 0.05 and were thus significantly associated with CNS toxicity: LCP1 (Lymphocyte Cytosolic Protein 1), q = 0.013, RETSAT (Retinol Saturase), q = 0.013 and SFMBT2 (Scm Like With Four Mbt Domains 2), q = 0.035 (Fig 2). The 20 genes with lowest q-values are shown in S3 Table.

Uncommon variants (VF < 0.123) within 140 bases from the closest exon/3'-UTR or 5'-UTR were included in the SKAT-O test. The number of included uncommon variants was 17 for LCP1, 26 for RETSAT, and 63 for SFMBT2. These 106 variants were individually tested in the same manner as in the common variant analysis (Fig 3). The Bonferroni corrected

**Table 2. Characteristics of the 66 included cases of central nervous system toxicity during treatment with antimicrobial drugs, and of the 833 controls.**

|  | Cases | Controls |
| --- | --- | --- |
| Sex (male/female) | 18/48 | 297/532 |
| Mean age (years [range])* | 46 [16–80] | 54 [2–89] |
| Suspected drugs† | Mefloquine (n = 13)<br>Ciprofloxacin (n = 5)<br>Trimethoprim (n = 5)<br>Atovaquone (n = 4)<br>Gentamicin (n = 4)<br>Ribavirin (n = 4)<br>Doxycycline (n = 3)<br>Metronidazole (n = 3)<br>Proguanil (n = 3)<br>Terbinafine (n = 3)<br>Tetracycline (n = 3)<br>Azithromycin (n = 2)<br>Clarithromycin (n = 2)<br>Nitrofurantoin (n = 2)<br>Peginterferon alpha-2a (n = 2)<br>Peginterferon alpha-2b (n = 2)<br>Tobramycine (n = 2)<br>Abacavir (n = 1)<br>Aciclovir (n = 1)<br>Amikacin (n = 1)<br>Chloroquine phosphate (n = 1)<br>Clavulanic acid (n = 1)<br>Dolutegravir (n = 1)<br>Elvitegravir (n = 1)<br>Emtricitabine (n = 1)<br>Phenoxymethylpenicillin (n = 1)<br>Hepatitis A vaccine (n = 1)<br>Hydroxychloroquine (n = 1)<br>Interferon beta-1b (n = 1)<br>Levofloxacin (n = 1)<br>Lymecycline (n = 1)<br>Sulfamethoxazole (n = 1)<br>Valaciclovir (n = 1) | N/A |
| Indications for treatment with suspected drugs | Malaria prophylaxis (n = 17)<br>Urinary tract infection (n = 11)<br>Pneumonia/respiratory infection (n = 7)<br>Acne (n = 4)<br>Hepatitis C (n = 4)Onychomycosis (n = 3)<br>Sinusitis (n = 3)<br>HIV (n = 2)<br>Endocarditis (n = 2)<br>Post-operative infections (n = 2)<br>Other infections (n = 6)<br>Other (n = 2)<br>Unknown (n = 2) | N/A |

(*Continued*)

**Table 2.** (Continued)

| | Cases | Controls |
|---|---|---|
| Adverse reactions† | Anxiety/panic attack (n = 11)<br>Ageusia/dysgeusia (n = 9)<br>Hallucinations (n = 7)<br>Depression/dysphoria (n = 7)<br>Dizziness/disturbed balance (n = 6)<br>Encephalitis/meningitis/myelitis (n = 6)<br>Headache (n = 5)<br>Impaired hearing (n = 5)<br>Nightmares (n = 5)<br>Anosmia (n = 4)<br>Intracranial pressure increased (n = 4)<br>Seizures (n = 3)<br>Insomnia (n = 3)<br>Difficulty concentrating (n = 3)<br>Confusion (n = 3)<br>Vestibular damage (n = 3)<br>Dysfasia (n = 2)<br>Psychosis (n = 2)<br>Suicidal ideation (n = 2)<br>Diplopia (n = 1)<br>Facial paralysis (n = 1)<br>Mood disorder (n = 1)<br>Persistent crying (n = 1)<br>Impaired consciousness (n = 1)<br>Migraine (n = 1)<br>Neck pain, sensory disturbance and paresthesia (n = 1)<br>Suicide attempt (n = 1)<br>Impaired vision (n = 1)<br>Delusions (n = 1) | Allergic reaction (n = 256)<br>Cytopenia (n = 136)<br>Liver toxicity (n = 105)<br>Other skin hypersensitivity (n = 86)<br>Atypical femoral fracture (n = 53)<br>Severe cutaneous adverse reaction (n = 51)<br>Bleeding (n = 49)<br>Phototoxicity (n = 45)<br>Pancreatitis (n = 39)<br>Renal toxicity (n = 33)<br>Tendon rupture (n = 22)<br>Hyponatremia/SIADH (n = 15)<br>Torsades de pointes/QT-prolongation (n = 15)<br>Weight gain (n = 15)<br>Pustulosis palmoplantaris (n = 13)<br>Psoriasis (n = 9)<br>Serum sickness (n = 2) |
| Concomitant diseases‡ | Hypertension (n = 14)<br>Osteoarthritis (n = 5)<br>Asthma (n = 5)<br>Coronary artery disease (n = 4) | Hypertension (n = 264)<br>Osteoarthritis (n = 184)<br>Rheumatic disease (n = 136)<br>Gallstone/cholecystitis (n = 110)<br>Infectious disease (n = 93)<br>Asthma (n = 84)<br>Cancer (n = 79)<br>Arrhythmia (n = 74)<br>Diabetes (n = 73)<br>Thyroid disease (n = 72)<br>Inflammatory bowel disease (n = 70)<br>Osteoporosis (n = 66)<br>Hyperlipidemia (n = 63)<br>Psoriasis (n = 62)<br>Coronary artery disease (n = 61)<br>Gastric/duodenal ulcer (n = 58)<br>Depression (n = 48)<br>Kidney stone (n = 46)<br>Epilepsy (n = 45) |

* Age was unknown for one case. For controls, values are based on 829 patients, with missing data for 4.

† Suspected drugs and adverse reactions are more than the total number of patients as several drugs were suspected in some cases, and patients could have several reported reactions.

‡ Showing concomitant diseases that were present in over 5% of the cohort.

significance threshold was the number of genes tested (N = 19250) added to the number of variants tested in the three genes (N = 106), resulting in p = 0.05/19356 = 2.58 x $10^{-6}$.

Two LCP1 variants in high linkage disequilibrium, rs6561297 and rs10492451, were associated with CNS toxicity (p = 1.15 x $10^{-6}$, VF cases = 0.121, VF controls = 0.032, OR: 4.60 [95%

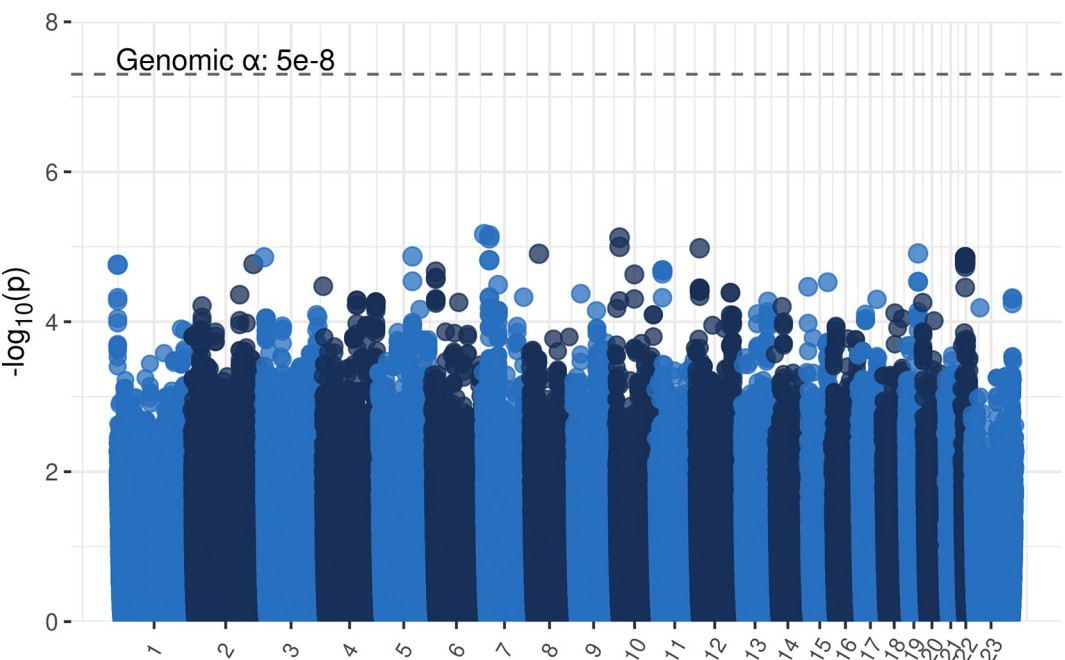

**Fig 1. Common variants (frequency $\geq$ 0.123) over the whole genome in cases with CNS toxicity (n = 66) vs controls (n = 833).** Tests were performed with logistic regression using PLINK2.0 and contained principal components one to four as covariates. The genome-wide significance threshold p = 5 x $10^{-8}$ (5e-8, dotted line) was used.

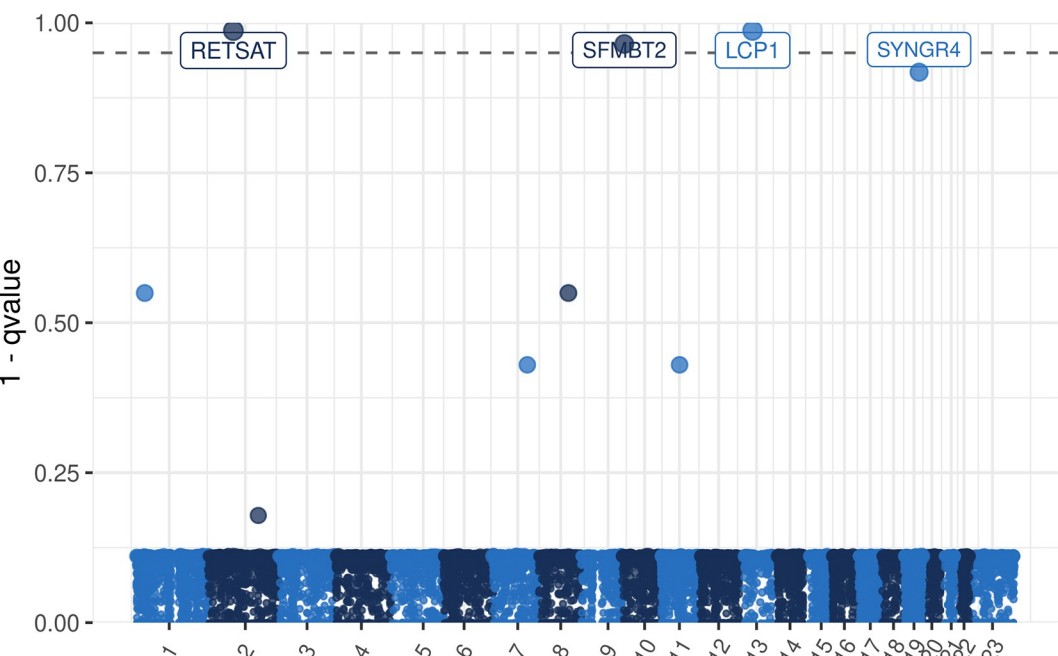

**Fig 2. Gene association tested using SKAT-O in cases with CNS toxicity (n = 66) vs controls (n = 833).** The number of genes tested was 19250. Uncommon variants (frequency < 0.123) within 140 bases from the closest exon/3'-UTR or 5'-UTR were included. The dotted line represents the significance threshold q-value < 0.05.

CI: 2.51–8.46], and p = 1.15 x $10^{-6}$, VF cases = 0.121, VF controls = 0.032, OR: 4.60 [95% CI: 2.51–8.46], respectively). Both are intronic variants, with rs10492451 being positioned a distal enhancer-like signature (EH38E1673884, listed in SCREEN V2) that is linked to LCP1.

No single variant was below the significance threshold for RETSAT. The top variants observed were rs60340620 (3-´ UTR, p = 2.107 x $10^{-5}$), rs139043592 (missense variant, p = 4.2739 x $10^{-5}$), rs145057327 (3-´ UTR, p = 4.81 x $10^{-5}$), rs572423656 (3-´ UTR, p = 4.81 x $10^{-5}$), rs192199548 (3-´UTR, p = 4.81 x $10^{-5}$), rs183409558 (3-´ UTR, p = 4.81 x $10^{-5}$) and rs143283662 (missense variant, p = 1.04 x $10^{-4}$).

No SFMBT2 variants clearly drove the association, but among the 63 variants tested, several were nonsynonymous. Additional frequencies, OR and p-values for the top 40 of the 106 variants can be found in S4 Table

### 3.4 Structural variants

Filtering out any genes that had less than six carriers among cases and controls resulted in 9775 genes (S5 Fig). Of these genes, 1255 were completely separated and retested using Chi-squared test. No association with CNS toxicity below the adjusted significance threshold of p < 5.12 x $10^{-6}$ (e-5.29) was detected, either using logistic regression or Chi-squared test.

### 3.5 HLA association

No notable association with HLA was observed below the adjusted significance threshold p < 1.62 x $10^{-4}$ (e-3.79), see S6 Fig.

## 4. Discussion

We investigated whether genetic variation in drug transporters expressed at the blood-brain barrier was associated with risk of CNS toxicity due to antimicrobial medications. No association was observed either with common genetic variation (frequency ≥ 0.123), uncommon variants (frequency < 0.123), or with structural variation in these genes.

We further explored whether any genetic variation across the whole genome was associated with risk of CNS toxicity due to antimicrobial drugs. Three genes were significantly associated with risk of CNS toxicity when testing uncommon variants with SKAT-O; LCP1, RETSAT and SFMBT2. Two intronic variants in LCP1 were identified to be driving the association, rs6561297 and rs10492451. One of them, rs10492451, is positioned in a distal enhancer-like signature, which could indicate functionality. It is important to note that only sufficiently common variants have the potential to survive the strict significance threshold, when tested as single variants. A function of L-Plastin (expressed by LCP1) is related to transport of T cell activation modules to the cell surface [35]. LCP1 is reported to be differentially expressed in brain tissue of people who died of suicide compared with accidental death [36], and in blood of sleep deprived individuals compared with well-rested controls [37]. A rare LCP1 missense variant has also been associated with schizophrenia in the Ashkenazi Jewish population [38]. However, none of these studies speculates on the possible role of LCP1 for these conditions.

Among the more common variants included in the SKAT-O test of RETSAT, two missense variants of (rs143283662 and rs139043592) were observed. RETSAT encodes retinol saturase that is involved in the metabolism of vitamin A, but the biological function of the enzyme product is unknown [39]. However, there is evidence for dysregulation of vitamin A metabolism in the etiology of schizophrenia [40].

SFMBT2 encodes the protein Scm-like with four malignant brain tumor domains 2. SFMBT2 has been reported to have a microglial transcriptional signature, to be differentially

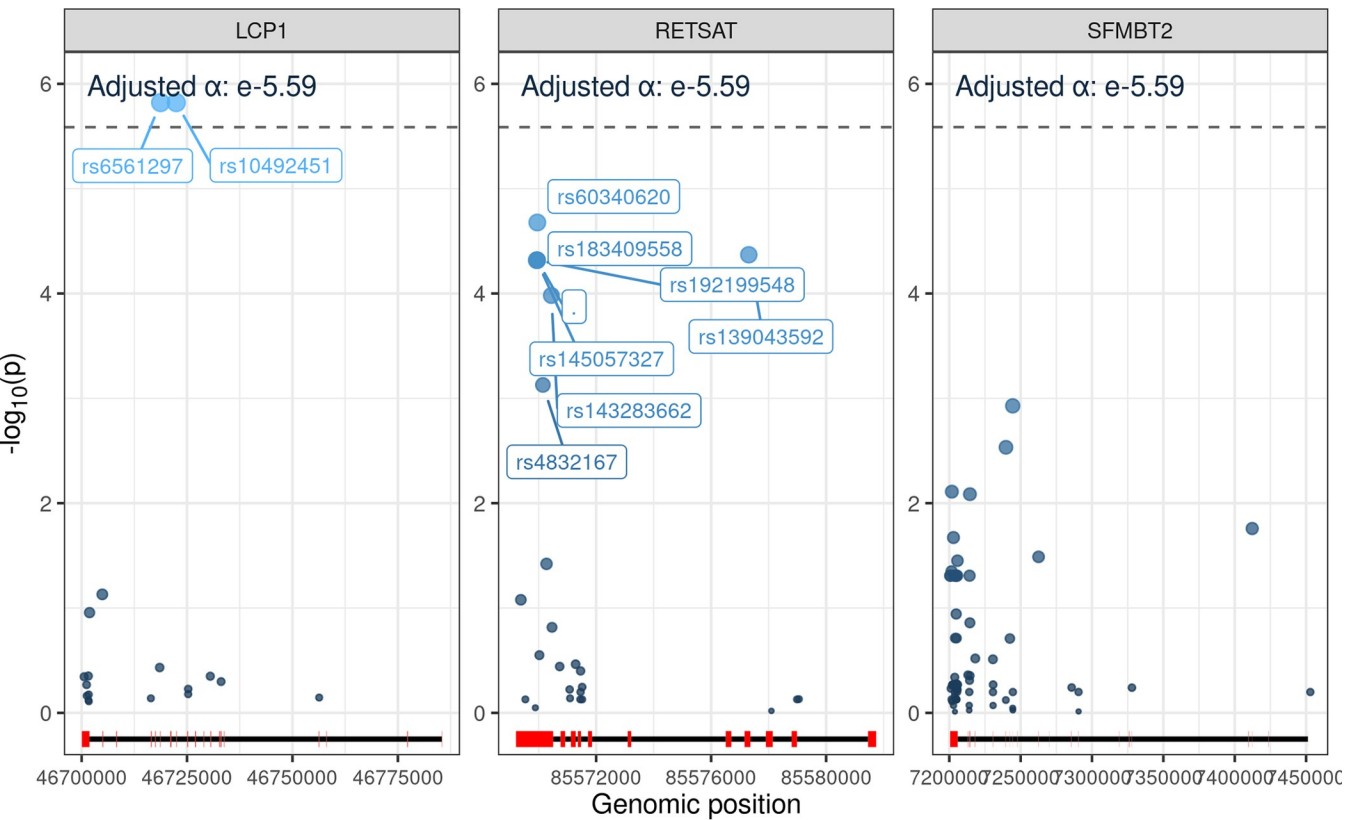

**Fig 3. Variants associated in gene-based tests using SKAT-O in cases with CNS toxicity (n = 66) vs controls (n = 833).** Each variant was tested using logistic regression with principal components one to four, calculated on all genetic variation, as covariates. The dotted line represents a Bonferroni corrected significance threshold. Exons marked in red and introns in black at the bottom of each graph.

expressed in Parkinson's Disease, and has been suggested to be involved in dementia with Lewy bodies [41]. However, no association between SFMBT2 and CNS toxicity due to drugs has been reported.

## 4.1 Limitations and strengths

The main limitation of this hypothesis generating study is the diversity of the drugs and adverse reactions examined, which could potentially mask genetic associations. Additionally, the small size of the cohort limits the detection of weaker genetic risk factors. However, due to the rarity of CNS toxicity induced by antimicrobial agents and its wide spectrum of manifestations, it is challenging to gather large cohorts.

To overcome these limitations, future studies should aim to gather larger and/or more homogeneous cohorts of patients with specific drug-reaction combinations. Such studies may require multi-center collaborations to obtain sufficient sample sizes and allow for meaningful conclusions to be drawn regarding the relationship between genetic variation and adverse drug reactions.

Nevertheless, it is worth noting that this study, to our knowledge, presents the largest curated cohort of whole genome sequenced patients with antimicrobial-induced CNS reactions to date which can be used for meta-analysis and targeted sequencing studies of the associations reported.

## 5. Conclusions

While this study represents an important step towards understanding the genetic basis of antimicrobial-induced CNS toxicity, further research is necessary to elucidate the full spectrum of genetic risk factors and their contributions to diverse adverse drug reactions. In our study, CNS toxicity due to antimicrobial drugs was associated with uncommon variation in LCP1, RETSAT and SFMBT2. However, these findings need to be replicated in studies with larger cohorts and/or that are more homogeneous.

## Supporting information

**S1 Fig. Principal component analysis plot.**
(DOCX)

**S2 Fig. Association p values of candidate gene variants.**
(DOCX)

**S3 Fig. Association p values of SKAT-O gene tests.**
(DOCX)

**S4 Fig. Association p values of structural variation.**
(DOCX)

**S5 Fig. Dominant test for structural variation.**
(DOCX)

**S6 Fig. Four-digit HLA type association test.**
(DOCX)

**S1 Table. Published data on the role of drug transporters.**
(DOCX)

**S2 Table. The 50 common variants with smallest p-value.**
(DOCX)

**S3 Table. Top 20 genes in SKATO test.**
(DOCX)

**S4 Table. Top variants present in genes significant in SKAT-O test.**
(DOCX)

## Acknowledgments

We are grateful to all physicians, research nurses, and supporting staff, who assisted in recruiting patients and controls or administering phenotype databases. In particular, we thank Associate Professor Marco Cavalli, Uppsala University, Sweden, for giving insights into functional genomics.

Whole genome sequencing of the patient cohort was performed at the single nucleotide polymorphism and sequencing (SNP&SEQ) Technology Platform at Uppsala University (www.genotyping.se) that is part of the National Genomics Infrastructure, Sweden. Computations were done on resources provided by the Swedish National Infrastructure for Computing through the Uppsala Multidisciplinary Centre for Advanced Computational Science (UPPMAX).

## Author Contributions

**Conceptualization:** Joel Ås, Henrik Green, Mia Wadelius, Pär Hallberg.

**Data curation:** Joel Ås, Ilma Bertulyte, Nina Norgren, Anna Johansson, Mia Wadelius, Pär Hallberg.

**Formal analysis:** Joel Ås, Nina Norgren, Anna Johansson, Niclas Eriksson.

**Funding acquisition:** Henrik Green, Mia Wadelius, Pär Hallberg.

**Investigation:** Ilma Bertulyte, Mia Wadelius, Pär Hallberg.

**Methodology:** Joel Ås, Nina Norgren, Anna Johansson.

**Project administration:** Mia Wadelius, Pär Hallberg.

**Software:** Joel Ås, Nina Norgren, Anna Johansson.

**Supervision:** Mia Wadelius, Pär Hallberg.

**Validation:** Ilma Bertulyte, Mia Wadelius, Pär Hallberg.

**Visualization:** Joel Ås.

**Writing – original draft:** Joel Ås, Ilma Bertulyte, Mia Wadelius, Pär Hallberg.

**Writing – review & editing:** Joel Ås, Ilma Bertulyte, Nina Norgren, Anna Johansson, Niclas Eriksson, Henrik Green, Mia Wadelius, Pär Hallberg.

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
