## [Decision Letter · Decision Letter 0]

1 Dec 2023

PONE-D-23-09336Whole genome study of central nervous system (CNS) toxicity due to antimicrobial drugsPLOS ONE

Dear Dr. Wadelius,

Thank you for submitting your manuscript to PLOS ONE. After careful consideration, we feel that it has merit but does not fully meet PLOS ONE’s publication criteria as it currently stands. Therefore, we invite you to submit a revised version of the manuscript that addresses the points raised during the review process.

We look forward to receiving your revised manuscript.

Kind regards,

Nur Aizati Athirah Daud, Ph.D.

Academic Editor

PLOS ONE

Journal Requirements:

"This study was granted by the Science for Life Laboratory’s Swedish Genomes Program 2017 that was supported by the Knut and Alice Wallenberg Foundation (application ID NP:00085). Furthermore, we received grants from the Swedish Research Council (Medicine 521-2011-2440, 521-2014-3370 and 2018-03307), and Clinical Research Support (Avtal om Läkarutbildning och Forskning, ALF) at Uppsala University. All grants mentioned were granted to MW and PH."

"The authors declare no conflict of interest. No funder took part in the study design, recruitment, analysis, interpretation of data, writing of the report or in the decision to submit the paper for publication. There are no other conflicts of interest."

Reviewers' comments:

Reviewer's Responses to Questions

**Comments to the Author**

1. Is the manuscript technically sound, and do the data support the conclusions?

Reviewer #1: Yes

Reviewer #2: Partly

2. Has the statistical analysis been performed appropriately and rigorously? 

Reviewer #1: Yes

Reviewer #2: No

3. Have the authors made all data underlying the findings in their manuscript fully available?

Reviewer #1: Yes

Reviewer #2: Yes

4. Is the manuscript presented in an intelligible fashion and written in standard English?

Reviewer #1: Yes

Reviewer #2: Yes

5. Review Comments to the Author

Reviewer #1: General comments:

Thank you for giving me the opportunity to review this manuscript. This paper is interesting because data that are important towards understanding the genetic basis of antimicrobial-induced CNS toxicity are still lacking.

Specific comments:

Good manuscript writing. Authors may refer to article “STrengthening the REporting of Genetic Association Studies (STREGA)— An Extension of the STROBE Statement” to improve the writing. However,

Title and Abstract:

Please indicate the study’s design with a commonly used term in the title or the abstract.

Materials and Methods:

Study design: Please present key elements of study design early in the paper.

Participants: Give the eligibility criteria (e.g. no exclusion criteria) and methods of selection of participants (sampling methods)

Variables: Clearly define all outcomes, exposures, predictors, potential confounders, and effect modifiers

Data sources measurement:

• Describe laboratory methods, including storage of DNA.

• State the laboratory/centre where genotyping was done.

• Describe comparability of laboratory methods if there is more than one group.

• Specify whether genotypes were assigned using all of the data from the study simultaneously or in smaller batches.

Bias: Describe any efforts to address potential sources of bias.

Study size: Explain how the study size was arrived at.

Statistical methods:

• Explain how missing data were addressed.

• State whether Hardy-Weinberg equilibrium was considered and, if so, how.

• Describe any methods used to address and correct for relatedness among subjects

Results:

Participants:

Report numbers of individuals in whom genotyping was attempted and numbers of individuals in whom genotyping was successful.

Notes: I am not able to access the supplementary files (tables/figures)

Reviewer #2: This paper explored whether any genetic variation across the whole genome was associated with risk of CNS toxicity due to antimicrobial drugs. Whole genome sequencing of 66 cases and 833 controls was performed. According to SKAT-O results, authors conclude CNS toxicity due to antimicrobial drugs was associated with uncommon variants in LCP1, RETSAT and SFMBT2.

There are some problems, which must be solved before the paper is considered for publication.

First, to apply rigorous quality control (QC) procedures. For example, in PCA figure (S1) some of the Swedegene dots are decentralized and some are even closer to the CHB and JPT populations, which imply the data need to further analyze. Sequencing quality or depth difference of case/control group in the positive regions should also be examined.

Second, considering the small sample size (especially case group), all negative results of association analysis need to be reduced in length (due to limited power).

Third, positive results in LCP1, RETSAT and SFMBT2 gene mainly relies on SKAT-O analysis. The conclusion is not solid

unless more evidence is added (such as replication in another population, population stratifications/subgroups analsyis , gene set analysis or gene function evidence).

Some other minor issues:

Line 277-278, p value and VF of two variants are exactly the same (inconsistency with description elsewhere in the text ).

Uncommon variants frequency (define as VF < 0.123 in the paper) is need for clearer rationale and explanation.

6. PLOS authors have the option to publish the peer review history of their article (what does this mean?). If published, this will include your full peer review and any attached files.

Reviewer #1: **Yes: **Zalina Zahari

Reviewer #2: **Yes: **Jian Guo

---

## [Author Response · Author response to Decision Letter 0]

31 Dec 2023

PONE-D-23-09336

Whole genome case-control study of central nervous system (CNS) toxicity due to antimicrobial drugs

Journal Requirements:

Reply: We have now formatted the manuscript according to PLOS ONE's style requirements.

"This study was granted by the Science for Life Laboratory’s Swedish Genomes Program 2017 that was supported by the Knut and Alice Wallenberg Foundation (application ID NP:00085). Furthermore, we received grants from the Swedish Research Council (Medicine 521-2011-2440, 521-2014-3370 and 2018-03307), and Clinical Research Support (Avtal om Läkarutbildning och Forskning, ALF) at Uppsala University. All grants mentioned were granted to MW and PH."

Reply: The role of the funder has been added to the financial disclosure on page 23 (in the version with tracked changes) and to the cover letter. "The funders had no role in study design, data collection and analysis, decision to publish, or preparation of the manuscript."

"The authors declare no conflict of interest. No funder took part in the study design, recruitment, analysis, interpretation of data, writing of the report or in the decision to submit the paper for publication. There are no other conflicts of interest."

Reply: Conditions for data sharing have been moved from the Data Availability Statement to the Competing Interests statement on page 23, and are also written in the cover letter. “Genetic data will be available at the official Federated European Genome-phenome Archive (FEGA) at NBIS/Elixir (https://fega.nbis.se/). Data access can be requested if the recipient adheres to our ethics approval, and the request is in accordance with FAIR data management and the General Data Protection Regulation (GDPR) 2016/679.”

Reply: The ethics statement has been moved from the Informed Consent statement at the end of the manuscript to the Methods section. See 2.1 on page 5 “The study was conducted according to the guidelines of the Declaration of Helsinki, and the study was approved by the regional ethical review board in Uppsala, Sweden (2010/231 Uppsala). Written informed consent was obtained from all participants in the study.”

Reply: Data will be made available as described above. As the minimal data consists of sensitive data the data cannot be shared without recipient adhering to our ethics approval, and GDPR. For this, an individual data management access approval contract needs to be signed.

Reviewers' comments:

Reviewer's Responses to Questions

Comments to the Author

1. Is the manuscript technically sound, and do the data support the conclusions?

Reviewer #1: Yes

Reviewer #2: Partly

2. Has the statistical analysis been performed appropriately and rigorously? 

Reviewer #1: Yes

Reviewer #2: No

3. Have the authors made all data underlying the findings in their manuscript fully available?

Reviewer #1: Yes

Reviewer #2: Yes

4. Is the manuscript presented in an intelligible fashion and written in standard English?

Reviewer #1: Yes

Reviewer #2: Yes

5. Review Comments to the Author

Reviewer #1: General comments:

Thank you for giving me the opportunity to review this manuscript. This paper is interesting because data that are important towards understanding the genetic basis of antimicrobial-induced CNS toxicity are still lacking.

Specific comments:

Good manuscript writing. Authors may refer to article “STrengthening the REporting of Genetic Association Studies (STREGA)— An Extension of the STROBE Statement” to improve the writing. 

Reply: We have completed the STROBE form.

However,

Title and Abstract:

Please indicate the study’s design with a commonly used term in the title or the abstract.

Reply: Thank you for pointing this out. The word “case-control” has been added to the title. 

Materials and Methods:

Study design: Please present key elements of study design early in the paper.

Reply: The study design is presented in the title and elaborated on in the abstract. Aims are reiterated in the last paragraph of the introduction (pages 3-4) and we have added a starting paragraph to the materials and methods (page 5) describing key elements of the study design. 

Participants: Give the eligibility criteria (e.g. no exclusion criteria) and methods of selection of participants (sampling methods)

Reply: Patients with a reported suspected CNS-toxic ADR were eligible, which has been added to page 5. Patients needed to be at least 18 years of age at the time of recruitment. Each patient was adjudicated and any patient with a suspected peripheral neurotoxicity was excluded (3.1 patient characteristics, page 12).

Variables: Clearly define all outcomes, exposures, predictors, potential confounders, and effect modifiers 

Reply: 

• Exposure: 

o Treatment with antimicrobial drugs

• Main Outcome and Measures:

o Presence of a CNS-toxic adverse drug reaction

o Correlation of genetic variation between cases and controls

• Effect modifiers:

o As the controls are not matched for drug treatment, we cannot exclude the possibility of some being at risk for CNS-toxic reactions. If there is a causal genetic predisposition this possibility could underestimate the observed correlations. This means that the correlations observed can be viewed as conservative.

• Potential confounders:

o With unmatched controls, the possibility of cofounding by indication exists. However, as the indications of cases are different types of infections or the preventive treatment for malaria, we do not believe that there is a genetic predisposition for these indications.

An overview of all drugs and CNS-toxic reactions among cases can be found in Table 2.

Data sources measurement:

• Describe laboratory methods, including storage of DNA. 

Reply: Laboratory methods are described on page 7. DNA storage has been added to page 6.

• State the laboratory/centre where genotyping was done.

Reply: The laboratory is described on page 6.

• Describe comparability of laboratory methods if there is more than one group.

Reply: Only one laboratory was used.

• Specify whether genotypes were assigned using all of the data from the study simultaneously or in smaller batches.

Reply: The full cohort wis sequenced in four batches. A clearer outline of these batches has been written under 2.4 on page 6.

Bias: Describe any efforts to address potential sources of bias

Reply: Samples with sequencing results of insufficient quality were re-sequenced. The cases were spread randomly throughout the batches. Individuals that were part of a study with strong known association to HLA-type were excluded (2.4, page 6). 

Study size: Explain how the study size was arrived at.

Reply: Attempts were made to include all available cases that fulfilled the criteria in Sweden 1990-2017. Recruitment was performed 2010-2017.

Statistical methods:

• Explain how missing data were addressed.

Reply: 

Missing data:

• Variant missingness: any variants with over 5% missing genotyping were excluded (2.7, page 8)

• Patient: NGI-U re-sequenced any failed samples (2.4, page 6)

• Parent’s country of birth was only used to validate PCA results. Outliers in genomic PCA were excluded (2.4, page 6). The current PCA plot (Fig S1) only shows the included cases and controls.

• State whether Hardy-Weinberg equilibrium was considered and, if so, how.

Reply: Hardy-Weinberg equilibrium:

o Any variant below the HWE threshold p < 10^-8 among controls was excluded (2.7, page 8)

• Describe any methods used to address and correct for relatedness among subjects

Reply: Methods used to address and correct for relatedness:

o The first 4 principal components were used as covariates in all tests performed.

Notes: I am not able to access the supplementary files (tables/figures)

Reviewer #2: This paper explored whether any genetic variation across the whole genome was associated with risk of CNS toxicity due to antimicrobial drugs. Whole genome sequencing of 66 cases and 833 controls was performed. According to SKAT-O results, authors conclude CNS toxicity due to antimicrobial drugs was associated with uncommon variants in LCP1, RETSAT and SFMBT2.

There are some problems, which must be solved before the paper is considered for publication.

First, to apply rigorous quality control (QC) procedures. For example, in PCA figure (S1) some of the Swedegene dots are decentralized and some are even closer to the CHB and JPT populations, which imply the data need to further analyze. Sequencing quality or depth difference of case/control group in the positive regions should also be examined. 

Reply: Thank you for your insightful comments. We took several measures that address your concerns relating to QC procedures in terms of sequence quality and sequencing depth deviation. We used multiple QC statistics and methods to look for inconsistencies between samples, regions and batches. NGI-U, who did the sequencing and initial pre-processing, delivered the data as recalibrated bam-files. NGI-U only released data that met their stringent quality criteria. In cases where samples did not meet these standards, they opted to re-sequence rather than provide subpar data. Due to this, there were no highly divergent outliers in terms of depth, coverage or sequencing quality. This is explained under Materials and Methods on pages 6-8. 

Thank you for pointing out the outlier in the population stratification plot (supplementary fig 1). In the previously submitted figure, it was not possible to discern whether the outlier was a case or a control. For enhanced clarity, we redrew the figure, and cases are now represented as squares and controls as circles. We agree that the single control that is closer to the CHB and JPT population could have been excluded. However, it is one of 833 controls, and its possible impact on the results is therefore negligible. We believe that this holds particularly true when examining the variants identified as driving the SKAT-O associations. Notably, these variants exhibit a significantly elevated variant frequency. It is essential to underscore that the maximum impact on variant frequency among controls is capped at +/- 1.2*10 ^-3. An important note here is that it could have been problematic if the outlier had been one of the cases, due to the higher influence this would have among the relatively few cases, where the largest effect would have been 1.5 * 10^-2. 

Second, considering the small sample size (especially case group), all negative results of association analysis need to be reduced in length (due to limited power).

Reply: The supplemental lists of SKAT-O and common variant analysis have been reduced by half in length (from 20 to 10, and from 50 to 25)

Third, positive results in LCP1, RETSAT and SFMBT2 gene mainly relies on SKAT-O analysis. The conclusion is not solid unless more evidence is added (such as replication in another population, population stratifications/subgroups analysis, gene set analysis or gene function evidence).

Reply:

• Replication in another population

o This is the long-term goal of this study, the problem with these ADRs is that they are rare, CNS-toxicity is heterogeneous, there is a multitude of antimicrobial drugs and in addition these ADRs are woefully underreported in healthcare. In order to expedite future research, any possible association found by our or another group needs to be communicated. Unfortunately, we do not know of any other material or studies that we could use for this today. 

• Subgroups analysis

o As the number of cases are small, we believe that increasing the complexity and decreasing the number of cases by performing a subgroup analysis would not be fruitful. While we would have liked to do this, we are of the opinion that a larger number of cases would have been required. 

• Gene set analysis

o Due to the heterogeneity of the question, we believe that it would be challenging to derive meaningful conclusions from enrichment tests. Analyzing previous studies that include findings related to LCP1, RETSAT, and SFMBT2, it appears that any direct relevance to our question is, perhaps, marginal. Consequently, combining these factors would likely result in an even more intricate outcome, making interpretation more difficult, in our opinion. 

• Gene function evidence

o The associated genes or their proteins, have previously been linked to symptoms that could be due to neurotoxicity, as discussed on pages 20-21.

Some other minor issues:

Line 277-278, p value and VF of two variants are exactly the same (inconsistency with description elsewhere in the text).

Reply: Thank you for bringing the discrepancy between the abstract and the results to our attention. The figures presented in the results and table S4 were correct. The two SNPs are in almost complete LD, and both SNPs have a variant frequency of 3% among cases. The abstract has now been updated (page 2).

Uncommon variants frequency (define as VF < 0.123 in the paper) is need for clearer rationale and explanation. 

Reply: We now describe how the variant frequency (VF) threshold was calculated using the Hardy-Weinberg principle with an expected value of one homozygote individual among cases, on page 9. “66 cases equate a threshold of 1/square root of 66 = 0.123.”

---

## [Decision Letter · Decision Letter 1]

5 Feb 2024

Whole genome case-control study of central nervous system (CNS) toxicity due to antimicrobial drugs

PONE-D-23-09336R1

Dear Dr. Wadelius,

We’re pleased to inform you that your manuscript has been judged scientifically suitable for publication and will be formally accepted for publication once it meets all outstanding technical requirements.

Kind regards,

Nur Aizati Athirah Daud, Ph.D.

Academic Editor

PLOS ONE

Additional Editor Comments (optional):

Reviewers' comments:

Reviewer's Responses to Questions

**Comments to the Author**

1. If the authors have adequately addressed your comments raised in a previous round of review and you feel that this manuscript is now acceptable for publication, you may indicate that here to bypass the “Comments to the Author” section, enter your conflict of interest statement in the “Confidential to Editor” section, and submit your "Accept" recommendation.

Reviewer #1: All comments have been addressed

Reviewer #2: All comments have been addressed

2. Is the manuscript technically sound, and do the data support the conclusions?

Reviewer #1: Yes

Reviewer #2: Partly

3. Has the statistical analysis been performed appropriately and rigorously? 

Reviewer #1: Yes

Reviewer #2: Yes

4. Have the authors made all data underlying the findings in their manuscript fully available?

Reviewer #1: Yes

Reviewer #2: Yes

5. Is the manuscript presented in an intelligible fashion and written in standard English?

Reviewer #1: Yes

Reviewer #2: Yes

6. Review Comments to the Author

Reviewer #1: Thank you for giving me the opportunity to review this manuscript and addressing all of my concerns with the original manuscript.

Reviewer #2: The authors adequately answered most of the questions. And the revised version meets the requirements for publication.

7. PLOS authors have the option to publish the peer review history of their article (what does this mean?). If published, this will include your full peer review and any attached files.

Reviewer #1: **Yes: **Zalina Zahari

Reviewer #2: **Yes: **Jian Guo

---

## [Editor Report · Acceptance letter]

15 Feb 2024

PONE-D-23-09336R1 

PLOS ONE

Dear Dr. Wadelius, 

I'm pleased to inform you that your manuscript has been deemed suitable for publication in PLOS ONE. Congratulations! Your manuscript is now being handed over to our production team.

Kind regards, 

on behalf of

Dr. Nur Aizati Athirah Daud 

Academic Editor

PLOS ONE